# Multidimensional Prognostic Index and Mortality in Intermediate Care Facilities: A Retrospective Study

**DOI:** 10.3390/jcm10122632

**Published:** 2021-06-15

**Authors:** Nicola Veronese, Stefano Vianello, Claudia Danesin, Florina Tudor, Gianfranco Pozzobon, Alberto Pilotto

**Affiliations:** 1Azienda ULSS (Unità Locale Socio Sanitaria) 3 “Serenissima”, 30174 Venice, Italy; stefano.vianello1@aulss3.veneto.it (S.V.); claudia.danesin@aulss3.veneto.it (C.D.); florina.tudor110890@gmail.com (F.T.); gianfranco.pozzobon@aulss3.veneto.it (G.P.); 2Geriatrics Section, Department of Medicine, University of Palermo, 90127 Palermo, Italy; 3Department Geriatric Care, Orthogeriatrics and Rehabilitation, Frailty Area, E.O. Galliera Hospital, 16128 Genova, Italy; alberto.pilotto@galliera.it; 4Department of Interdisciplinary Medicine, Aldo Moro University of Bari, 70121 Bari, Italy

**Keywords:** multidimensional prognostic index, mortality, prognosis, intermediate care

## Abstract

Multidimensional prognostic index (MPI) is a frailty assessment tool used for stratifying prognosis in older hospitalized people, but data regarding older people admitted to intermediate care facilities (ICFs) are missing. The aim of this study is to evaluate whether MPI can predict mortality in older patients admitted to the ICFs. MPI was calculated using different domains explored by a standard comprehensive geriatric assessment and categorized into tertiles (MPI-1 ≤ 0.20, MPI-2 0.20–0.34, MPI-3 > 0.34). A Cox’s regression analysis, taking mortality as the outcome, was used, reporting the results as hazard ratios (HRs) with 95% confidence intervals (CIs). In total, 653 older patients were enrolled (mean age: 82 years, 59.1% females). Patients in MPI-2 (HR = 3.66; 95%CI: 2.45–5.47) and MPI-3 (HR = 6.22; 95%CI: 4.22–9.16) experienced a higher risk of mortality, compared to MPI-1. The accuracy of MPI in predicting mortality was good (area under the curve (AUC) = 0.74, 95%CI: 0.70–0.78). In conclusion, our study showed that prognostic stratification, as assessed by the MPI, was associated with a significantly different risk of mortality in older patients admitted to the ICFs, indicating the necessity of using a CGA-based tool for better managing older people in this setting as well.

## 1. Introduction

Several older patients have an increasingly number of complex or specialized diseases and are consequently treated in hospitals [1]. However, the severity of these conditions (or the treatment requirements) do not always justify the hospitalization and, at the same time, may preclude discharge [1]. For these reasons, an increasing number of special units (namely intermediate care facilities, ICFs) are being set up to offer specialized treatments and close monitoring for older patients before a final discharge at home [1]. ICFs seem to also have relevant public health implications, since these units significantly reduce costs compared to hospital management.

However, several obstacles are present in the discharge from the ICFs to home [2]. This suggests the need of a precise stratification of older people, taking into account prognostic factors that should include functional, physical and psycho-social factors usually assessed by a geriatric multidimensional assessment [3]. This approach might help the physician in correctly managing the older patients admitted to the ICFs [3]. In this regard, the multidimensional prognostic index (MPI) is a prognostic tool based on a standard comprehensive geriatric assessment (CGA), which is used to address short- and long-term mortality risk, and was previously validated in hospitalized older patients [4]. Several studies have demonstrated that MPI has an excellent accuracy and calibration in predicting negative clinical outcomes (e.g., mortality, hospitalization, admission into nursing home) in a hospital setting [5,6]. Furthermore, the MPI has been validated in over 54,000 older adults suffering from the most common chronic and acute age-related diseases [7], and is thus considered one of the most commonly used tools to assess frailty in older people, not only in hospital, but also in other settings, such as primary care [8].

Unfortunately, very few data are available regarding the use and the clinical importance of prognostic tools for older people in ICFs, even if it is largely demonstrated that ICFs can significantly contribute to avoiding hospital admission, support early discharge, and enable the regaining of abilities in daily living for frail older people [9,10].

Given this background, the aim of this study is to evaluate if the MPI can predict mortality in older patients admitted to ICFs.

## 2. Materials and Methods

### 2.1. Participants

For the aims of this work, we considered all patients admitted to six ICFs for a total of 90 beds, between 1 March and 31 December 2020 in Venice, Italy. From 15 March 2020, the ICFs in Venice were allowed to host patients affected by COVID-19. During 2020, 69 patients with COVID-19, diagnosed using a molecular nasopharyngeal swab, were admitted to our ICFs.

The ICFs in Italy can guarantee intermediate care, defined as the care necessary for those patients who are medically stabilized (who do not require assistance in hospital, but are too unstable to be treated as outpatients or residential) and that treats problems that are solved in a limited period of time (usually no longer than 4 weeks) [11].

The study was approved by our local Ethical Committee on 24 November 2020.

### 2.2. Exposure: The Multidimensional Prognostic Index (MPI)

A version of the MPI, modified from the original version that originally included information regarding nutritional status, disability, number of medications, risk of pressure sores, severity of comorbidities, social aspects [4] and using tests commonly used for the admission to the ICFs, was used [12,13]. This tool requires about 15–20 min for trained health professionals [12].

In summary, nine domains, including 55 different questions, were included: (1) age; (2) sex; (3) main diagnosis; (4) nursing care needs (VIP); (5) cognitive status (VCOG), evaluated by the Short Portable Mental Status Questionnaire (SPMSQ) [14]; (6) pressure sores risk (VPIA), evaluated by the Exton–Smith scale [15]; (7) activities of daily living (VADL) and (8) mobility (VMOB) evaluated by the Barthel Index [16]; and (9) social support (VSOC) [12]. To calculate this MPI, a weighted sum of each individual domain, taking as the outcome mortality after one month, was used [12]. The MPI is routinely calculated using a standardized Microsoft Excel sheet.

The participants were divided, for statistical reasons, according to two MPI cut-offs into tertiles, i.e., 0.20 and 0.34, considering MPI-1 the lowest tertile (MPI 0–0.20), MPI-2 the middle tertile (MPI 0.20–0.34), and MPI-3 the highest tertile (MPI > 0.34).

### 2.3. Outcome: Mortality

The primary outcome of our research was mortality. The data regarding mortality are collected routinely as administrative data.

### 2.4. Statistical Analysis

Continuous variables were evaluated in terms of means and standard deviation (SD), and were categorically relative in terms of frequencies (%). Levene’s test was used to test the homoscedasticity of variances and, if its assumption was violated, Welch’s ANOVA was used. *p* values for trends were calculated using the Jonckheere–Terpstra test for continuous variables and the Mantel–Haenszel Chi-square test for categorical ones.

The association between MPI and mortality was made using different approaches. First, we reported the incidence of the outcome of interest, per 1000 person-days, by MPI tertiles. Second, we applied the log-rank test, using MPI in tertiles. Moreover, we assessed the association of MPI with mortality using a Cox’s regression analysis. The results were consequently reported as hazard ratios (HRs) with their 95% confidence intervals (95%CI). Similar analyses were run, using MPI, with increases of 0.10 points. The accuracy of MPI was evaluated with the 5-fold cross-validated area under the curve (AUC), with the correspondent 95%CIs.

All analyses were performed using the SPSS 21.0 for Windows (SPSS Inc., Chicago, IL, USA). All statistical tests were two-tailed and statistical significance was assumed for a *p*-value < 0.05.

## 3. Results

### 3.1. Sample Selection

Among 700 ICFs patients, 47 were aged less than 60 years and, consequently, were excluded from this study, leaving 653 patients for our analyses. These patients had a mean age of 82 ± 6 years (range: 60–100) and were mainly females (59.1%).

### 3.2. Baseline Characteristics

Table 1 shows the baseline characteristics by MPI tertiles measured at the baseline. Compared to those in the MPI-1 group, people in MPI-3 were significantly older, had a greater prevalence of dementia and immobilization syndrome, and ultimately showed worse values in all domains enclosed in the MPI, including activities of daily living, cognitive function, nursing care needs, mobility, pressure sores risk, and social support network (Table 1).

### 3.3. Mortality Data

During the follow-up period of 365 days, we recorded 220 deaths, with a global incidence of 3.42 (2.99–3.91) per 1000 person-days. Table 2 shows the mortality data according to the MPI values. The incidence rate of mortality in MPI-3 was about eight times more than in MPI-1 (7.94 vs. 1.04 per 1000 person-days, *p* < 0.0001) (log-rank test < 0.0001) (Figure 1).

Taking those in MPI-1 as the reference group, patients in MPI-2 (HR = 3.66; 95%CI: 2.45–5.47; *p* < 0.0001) and MPI-3 (HR = 6.22; 95%CI: 4.22–9.16; *p* < 0.0001) reported a significantly higher risk of death. Similar results were evident when modeling MPI as the continuous variable, since an increase of 0.10 points was associated with an increased risk of death of 43% (HR = 1.43; 95%CI: 1.34–1.52; *p* < 0.0001) (Table 2).

Finally, as shown in Figure 2**,** MPI has a good accuracy in predicting mortality in older patients admitted to the ICFs, with an AUC = 0.74 (95%CI: 0.70–0.78).

## 4. Discussion

In this study, including 653 older patients admitted to ICFs, higher MPI values were associated with a significantly higher risk of mortality. In this regard, the accuracy of MPI in predicting death was good, with an AUC of 0.74, which is close to that of previous studies performed with hospitalized older patients admitted for acute medical conditions [4,6]. To the best of our knowledge, this is the first study exploring the use of a prognostic tool such as MPI in ICFs.

A Delphi study [17] examining the models of the ICFs underlined that, with this term, we can include a broad range of time-limited services, ranging from crisis response to support over weeks to months that aims to ensure continuity, improve quality of care and promote recovery [18,19]. Even if transitional and intermediate care services may be limited to some weeks [20], they are more comprehensive than discharge planning activities or chronic care management [17]. In reality, for example, the ICFs have the main aim of attenuating discharge from hospital to home and, therefore, the maximum time is set as 4 weeks, which is extendable in the case of physician’s authorization.

The topic of the prognosis in older patients admitted to the ICFs is of importance, since clinical decision-making tools help the physician in managing older people in different settings [21]. As declared before, one of the most important aspects of the ICFs is that these structures have strict time limitations. Therefore, to know the prognosis of an older patient in the ICF can help the physician in better approaching his/her needs for discharge at home. Even if the same ICFs themselves seem to be effective in preventing and treating frailty in older people [22], unfortunately, no study was available regarding the use of prognostic tools in these structures, suggesting that this topic should be better explore in future research.

Therefore, we believe that our findings could be important for physicians working in the ICFs for several reasons. First, MPI in this setting is an accurate tool, indicating that it could be routinely used for better treating older people admitted into these units. In other contexts, MPI could be used for therapeutical [23] and intervention decisions [24], taking into account the fundamental role of prognosis. Second, to determine the prognosis of older people through an accurate tool can improve the clinical decision-making of the physicians, since MPI can precisely indicate older people needing hospitalization (in our case, those in the MPI-1 group) or other more conservative approaches (MPI-3 group), which these groups being stratified via mortality risk. Finally, MPI has already been used as a prognostic factor in settings other than ICFs during the COVID-19 epidemic. In particular, in nursing home residents, we have shown that higher MPI values indicate a higher risk of mortality, independently of the presence or not of COVID-19 and vice versa (i.e., COVID-19 increases the risk of mortality independently of MPI values) [13]. More recently, we have reported that in older patients hospitalized for COVID-19, MPI is useful for better stratifying the risk of mortality, whilst more data are needed for predicting the risk of ICU admission [25].

The findings of our study should be considered in the context of its limitations. First, there is the observational nature of this research. Second, we did not explore outcomes other than mortality, such as hospitalization or nursing home admission, which are, on the contrary, of public health importance. Other studies are therefore needed.

## 5. Conclusions

Our study showed that prognostic stratification, as assessed by the MPI, was associated with a significantly different risk of mortality in older patients admitted into the ICFs. We think that our findings are of importance for finally introducing prognostic factors, derived from CGA and the validated scales largely used in geriatric medicine, into this setting, which is rapidly proliferating around the world.

## Figures and Tables

**Figure 1 jcm-10-02632-f001:**
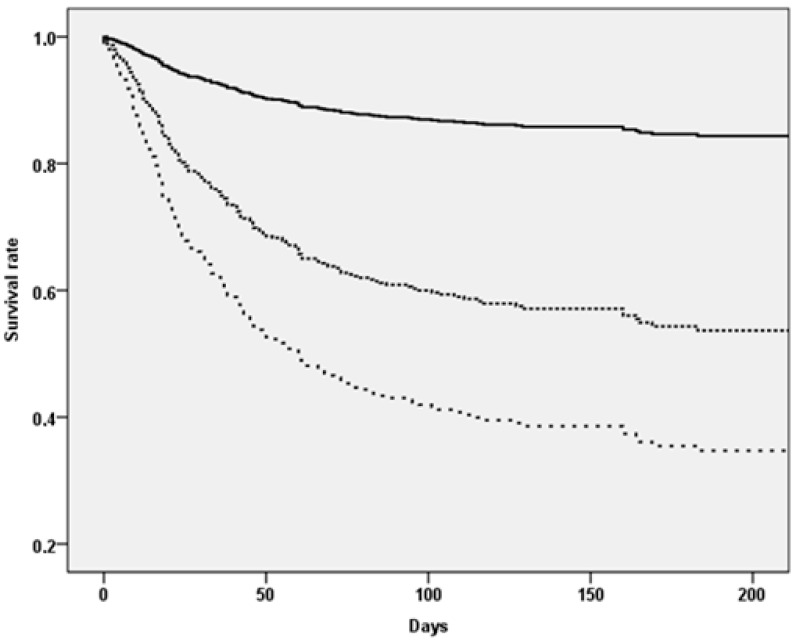
The upper continuous line represents people in MPI-1, the (intermediate) dashed line participants in MPI-2 and the bottom line those in MPI-3.

**Figure 2 jcm-10-02632-f002:**
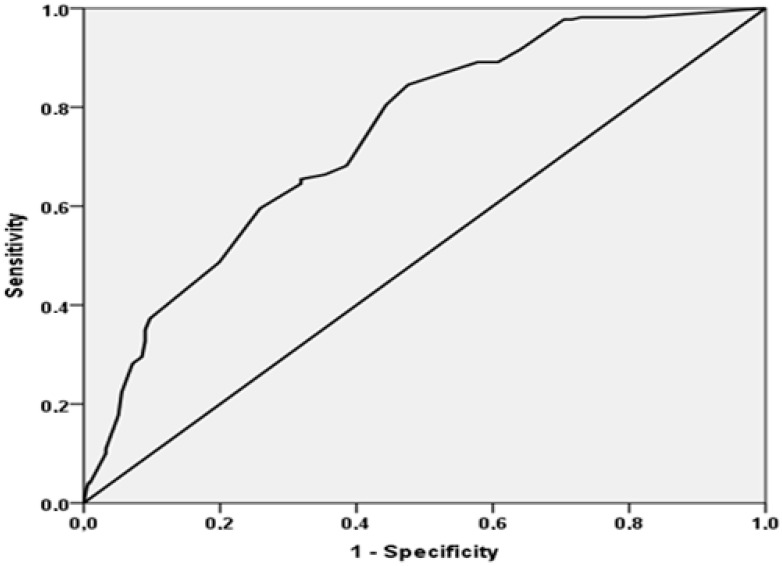
Area under the curve (AUC) of the MPI in predicting mortality.

**Table 1 jcm-10-02632-t001:** Descriptive analysis of intermediate care facility patients, by multidimensional prognostic index tertiles.

Domain	MPI-1 (*n* = 261)	MPI-2 (*n* = 199)	MPI-3 (*n* = 193)	*p*-Value
Age	81 (9)	81 (8)	84 (8)	0.001
Female sex (%)	60.5	57.8	58.5	0.82
Dementia (%)	3.8	3.6	26.1	<0.0001
Immobilization syndrome (%)	6.1	6.2	25.1	<0.0001
VIP	3.72 (5.00)	6.13 (5.93)	15.41 (10.99)	<0.0001
VPIA	1.46 (3.69)	6.46 (5.50)	8.19 (6.85)	<0.0001
VCOG	3.25 (3.06)	6.59 (3.45)	6.32 (3.20)	<0.0001
VADL	28.7 (18.0)	52.5 (13.7)	54.2 (9.4)	<0.0001
VMOB	25.4 (13.1)	35.5 (8.7)	38.0 (3.8)	<0.0001
VSOC	174 (57)	209 (43)	215 (44)	<0.0001
MPI	0.08 (0.08)	0.30 (0.04)	0.52 (0.13)	<0.0001

Abbreviations: MPI, multidimensional prognostic index; VADL, activities of daily living; VCOG, cognitive functions; VIP, nursing care needs; VMOB, mobility; VPIA, pressure sores risk; VSOC, social support network.

**Table 2 jcm-10-02632-t002:** Association between multidimensional prognostic index and mortality in intermediate care facility patients.

MPI Category	Number of Deaths	Number of Subjects	Incidence Rate (per 1000 Persons-Days) (95% CI)	Risk of Mortality (HR, 95%CI)	*p*-Value
MPI-1	34	261	1.04 (0.74–1.46)	1 [reference]	-
MPI -2	79	199	4.26 (3.41–5.33)	3.66 (2.45–5.47)	<0.0001
MPI-3	107	193	7.94 (6.57–9.62)	6.22 (4.22–9.16)	<0.0001
Increase in 0.10 points in MPI	-	-	-	1.43 (1.34–1.52)	<0.0001

Abbreviations: MPI, multidimensional prognostic index.

## Data Availability

Data are available upon request to the corresponding author.

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
