# Peer review of "Multidimensional Prognostic Index and Mortality in Intermediate Care Facilities: A Retrospective Study"

_jcm, 2021, doi:10.3390/jcm10122632_

Round 1

Reviewer 1 Report

I read with interest the article of Nicola Veronese et al. about Intermediate Care Facilities. This paper is easy to read and quite informative about prognosis among elderly patient.

I have some minor comments and questions.

Title :

the title precise “during pandemic COVID-19” but during my read, I don’t detect the extent to which the pandemia had an impact on the MPI value and use ? Perhaps a sentence about this aspect would be nice. If no impact/ change was noticed, the “during Pandemic COVID-19” should be removed from the title.

Introduction:

3 references are use in the introduction ( Nasraway et al., Hasin et al. and Vincent et al.). Theses ref deal with Intermediate care unit. There is, in my point of view, a semantic confusion. Indeed, in theses article, intermediate care unit indicate unit managing patients who need more care than a general ward can provide but do not really need the degree of monitoring and expertise that an ICU offers (JL Vincent). That’s quite different of “intermediate care Facilities” which take care patient after ward and before home return. Thank to clarify of remove theses references, in order to avoid any confusion (this confusion is often find in scientific medical literature).

Material and Methods

Line 70 : there is writte : “slightly modified”. Please precise which modifications have been done.

Line 80 : please explain how the cut-offs of MPI was chosen (previous publication, statistical reason ?)

The MPI seems quite difficult to fill, regarding the number of questions. Have clinician any electronic devices to do such score?

Figures / table

Legends of each table/figures should be reorganised. For instance, I don’t see very well the legend of KM curves, and its distinction with table 2 legends.

About Kaplan Meier curves: did the author performed a Logrank test ?

Discussion:

Line 177-178 : I don’t well understand this sentence :

“can improve the clinical decision making of the physicians since MPI can precisely indicate older people needing re-hospitalization (in our case those in MPI-1 group) or other more conservative approaches (MPI-3 group)”.

I didn’t see data about MPI and rehospitalisation. This statement is, in my sense, up to debate without data.

Author Response

Reviewer1

Title :

the title precise “during pandemic COVID-19” but during my read, I don’t detect the extent to which the pandemia had an impact on the MPI value and use ? Perhaps a sentence about this aspect would be nice. If no impact/ change was noticed, the “during Pandemic COVID- 19” should be removed from the title.

R: Fully agree with this observation. Even if the study was made during 2020, the topic of COVID-19 is practically not treated. Therefore, we removed from the title.

Introduction:

3 references are use in the introduction ( Nasraway et al., Hasin et al. and Vincent et al.). Theses ref deal with Intermediate care unit. There is, in my point of view, a semantic confusion. Indeed, in theses article, intermediate care unit indicate unit managing patients  who need more care than a general ward can provide but do not really need the degree of monitoring and expertise that an ICU offers (JL Vincent). That’s quite different of

“intermediate care Facilities” which take care patient after ward and before home return.  Thank to clarify of remove theses references, in order to avoid any confusion (this confusion is often find in scientific medical literature).

R: We are very sorry for this inconvenience and we sincerely thank the Reviewer for this correct observation. We have removed these references, replacing with one more appropriate (actual reference 1).

Material and Methods

Line 70 : there is writte : “slightly modified”. Please precise which modifications have been done.

R: We have better explained what we mean with the original MPI, as follows:

“A version of the MPI, modified from the original version that originally includes information re-garding nutritional status, disability, number of medications, risk of pressure sores, sever-ity of comorbidities, social aspects.”

Line 80 : please explain how the cut-offs of MPI was chosen (previous publication, statistical reason ?)

R: For this publication, we chose the tertiles for statistical reasons. Now, this is better explained at line 80.

The MPI seems quite difficult to fill, regarding the number of questions. Have clinician any electronic devices to do such score?

R: Good point. We have created a Microsoft Excel sheet that automatically calculated the MPI, having the information regarding the domains. We added this concept in the Methods section.

Figures / table

Legends of each table/figures should be reorganised. For instance, I don’t see very well the  legend of KM curves, and its distinction with table 2 legends.

R: We added a figure legend, after Figure 1 for better explaining the findings of the figure, as suggested.

About Kaplan Meier curves: did the author performed a Logrank test ?

R: Sorry for this inconvenience. We used the log-rank test in our analyses. Now we added this concept in the Methods section and in the Results.

Discussion:

Line 177-178 : I don’t well understand this sentence :

“can improve the clinical decision making of the physicians since MPI can precisely indicate older people needing re-hospitalization (in our case those in MPI-1 group) or other more conservative approaches (MPI-3 group)”.

I didn’t see data about MPI and rehospitalisation. This statement is, in my sense, up to debate without data.

R: We are very sorry for this inconvenience. We have better explained this passage, as follows:

“Second, to know the prognosis of older people through an accurate tool, can  improve the clinical decision making of the physicians since MPI can precisely indicate older people needing hospitalization (in our case those in MPI-1 group) or other more conservative approaches (MPI-3 group) in case of necessity, being stratified at different mortality risk.”

Reviewer 2 Report

Ms. No.: JCM-1237023-peer-review-v1

Title:  Multidimensional Prognostic Index and Mortality in Intermediate Care Facilities during Pandemic COVID-19: a Retrospective Study

Date:  26 May 2021

The manuscript “Multidimensional Prognostic Index and Mortality in Intermediate Care Facilities during Pandemic COVID-19: a Retrospective Study” is innovative, as the use of the tool has been poorly studied. The design and methodology are sound and the study includes a relatively large sample. The manuscript is well written. Minor changes have been suggested to authors, as follows:

  1. Introduction section: The study is conducted in patients with COVID-19, but the disease is hardly mentioned. The manuscript would benefit if a paragraph about the role of prognosis index within this population is included in the introduction section.
  2. Methods section.

2.1. Design: The manuscript might improve if authors state clearly the study design at the beginning of the methods section.

  • Population: Describe the procedures and criteria used for the diagnosis of COVID-19.
  • Follow-up: State the length of the follow-up in the methods section, please.
  1. Discussion section: If there are other studies that have applied this index in patients with COVID-19, it would be interesting to discuss their results. Does the index show different results in COVID-19 patients than in general older population?

Author Response

Reviewer 2

The manuscript “Multidimensional Prognostic Index and Mortality in Intermediate Care Facilities during Pandemic COVID-19: a Retrospective Study” is innovative, as the use of

the tool has been poorly studied. The design and methodology are sound and the study includes a relatively large sample. The manuscript is well written. Minor changes have been suggested to authors, as follows:

  1. Introduction section: The study is conducted in patients with COVID-19, but the disease is hardly mentioned. The manuscript would benefit if a paragraph about the role of prognosis index within this population is included in the introduction

R: Even if this study was made during the COVID-19 epidemic, the role of this condition is marginal. Therefore, we removed from the title and the manuscript this concept.

  1. Methods

  • Design: The manuscript might improve if authors state clearly the study design at the beginning of the methods section.

  • Population: Describe the procedures and criteria used for the diagnosis of COVID-

R: Please see our previous answer. However, we have reported a sentence regarding COVID-19 patients, as follows:

“During 2020, 69 patients with COVID-19, diagnoses using a molecular nasopharyngeal swab, were admitted in our ICFs.”

  • Follow-up: State the length of the follow-up in the methods section,

R: Added, as follows: “During the follow-up period of 365 days…”

  1. Discussion section: If there are other studies that have applied this index in patients with COVID-19, it would be interesting to discuss their results. Does the index show different results in COVID-19 patients than in general older population?

R: We sincerely thank the Reviewer for this question. In people affected by COVID-19, the MPI was already used. We have now added this paragraph as follows in the Discussion section:

“In particular, in nursing home residents we have shown that higher MPI values indicate a higher risk of mortality, independently from the presence or not of COVID-19 and vice versa (i.e., COVID-19 increases the risk of mortality independently from MPI values).[14] More recently, we have reported that in hospitalized older patients for COVID-19, MPI is useful for better stratifying the risk of mortality, whilst more data are needed for predicting the risk of ICU admission.[27]